# HIV Pretreatment Drug Resistance Trends in Mexico City, 2017–2020

**DOI:** 10.3390/pathogens10121587

**Published:** 2021-12-08

**Authors:** Claudia García-Morales, Daniela Tapia-Trejo, Margarita Matías-Florentino, Verónica Sonia Quiroz-Morales, Vanessa Dávila-Conn, Ángeles Beristain-Barreda, Miroslava Cárdenas-Sandoval, Manuel Becerril-Rodríguez, Patricia Iracheta-Hernández, Israel Macías-González, Rebecca García-Mendiola, Alejandro Guzmán-Carmona, Eduardo Zarza-Sánchez, Raúl Adrián Cruz, Andrea González-Rodríguez, Gustavo Reyes-Terán, Santiago Ávila-Ríos

**Affiliations:** 1Centre for Research in Infectious Diseases, National Institute of Respiratory Diseases, Calzada de Tlalpan 4502, Colonia Sección XVI, Mexico City 14080, Mexico; claudia.garcia@cieni.org.mx (C.G.-M.); daniela.tapia@cieni.org.mx (D.T.-T.); margarita.matias@cieni.org.mx (M.M.-F.); veronica.quiroz@cieni.org.mx (V.S.Q.-M.); vanessa.davila@cieni.org.mx (V.D.-C.); angeles.beristain@gmail.com (Á.B.-B.); sandovalmiroslava14@gmail.com (M.C.-S.); manuel.becerril@cieni.org.mx (M.B.-R.); eduardo.zarza@cieni.org.mx (E.Z.-S.); 2Condesa Specialised Clinic, General Benjamín Hill 24, Colonia Condesa, Mexico City 06140, Mexico; patricia.iracheta.hedz@gmail.com (P.I.-H.); maglezis@gmail.com (I.M.-G.); andrea.gonzalez.condesa@gmail.com (A.G.-R.); 3Condesa Iztapalapa Specialised Clinic, Combate de Celaya s/n, Colonia Unidad Habitacional Vicente Guerrero, Mexico City 09730, Mexico; rebe_ime@yahoo.com.mx (R.G.-M.); alex.guzman75@gmail.com (A.G.-C.); acruzf@gmail.com (R.A.C.); 4Coordinating Commission of the National Institutes of Health and High Specialty Hospitals, Periférico Sur 4809, Colonia Arenal de Tepepan, Mexico City 14610, Mexico; gustavo.reyesteran@gmail.com

**Keywords:** HIV pretreatment drug resistance, HIV acquired drug resistance, Mexico City, Mexico

## Abstract

In response to increasing pretreatment drug resistance (PDR), Mexico changed its national antiretroviral treatment (ART) policy, recommending and procuring second-generation integrase strand-transfer inhibitor (INSTI)-based regimens as preferred first-line options since 2019. We present a four-year observational study describing PDR trends across 2017–2020 at the largest HIV diagnosis and primary care center in Mexico City. A total of 6688 baseline protease-reverse transcriptase and 6709 integrase sequences were included. PDR to any drug class was 14.4% (95% CI, 13.6–15.3%). A significant increasing trend for efavirenz/nevirapine PDR was observed (10.3 to 13.6%, *p* = 0.02). No increase in PDR to second-generation INSTI was observed, remaining under 0.3% across the study period. PDR was strongly associated with prior exposure to ART (aOR: 2.9, 95% CI: 1.9–4.6, *p* < 0.0001). MSM had higher odds of PDR to efavirenz/nevirapine (aOR: 2.0, 95% CI: 1.0–3.7, *p* = 0.04), reflecting ongoing transmission of mutations such as K103NS and E138A. ART restarters showed higher representation of cisgender women and injectable drug users, higher age, and lower education level. PDR to dolutegravir/bictegravir remained low in Mexico City, although further surveillance is warranted given the short time of ART optimization. Our study identifies demographic characteristics of groups with higher risk of PDR and lost to follow-up, which may be useful to design differentiated interventions locally.

## 1. Introduction

Antiretroviral therapy (ART) has provided undeniable benefits at the individual and population levels, significantly reducing morbidity and mortality among people living with HIV (PLVIH) [1] and averting new infections [2]. An estimated 27.5 million PLHIV were on ART by the end of 2020 worldwide [3]. However, the widespread use of ART has been associated with the rise and spread of HIV drug resistance (HIVDR) [4]. According to the World Health Organization (WHO) operational definition, pretreatment drug resistance (PDR) refers to HIVDR detected in ART-naïve persons or previously antiretroviral (ARV)-exposed persons reinitiating first-line ART [5]. Over the last decade, there is growing evidence that PDR to non-nucleoside reverse transcriptase inhibitors (NNRTIs), mainly efavirenz and nevirapine, has been increasing in low- and middle-income countries (LMICs) [6]. Additionally, nationally representative surveys have been performed in several LMICs evidencing efavirenz/nevirapine PDR levels over 10% [4]. The high NNRTI PDR levels observed have led to the WHO recommendation and advocacy of the use of dolutegravir-based first-line regimens as the preferred option in LMICs [7].

In the Mexican context, two nationally representative surveys have shown efavirenz/nevirapine PDR levels close to 10% [8,9]. In addition, a study by our group demonstrated significant increases in efavirenz/nevirapine PDR levels in different areas of the country from 2008 to 2016, including Mexico City [10]. These results, together with advocacy and stewardship efforts, led to a change in national policy to recommend and procure ART regimens containing second-generation integrase strand-transfer inhibitors (INSTI), mainly bictegravir, as preferred first-line options since 2019 [11]. In Mexico, as in other LMICs, baseline HIVDR testing is not standard of care according to national guidelines.

To date, no data have been published updating PDR trends and describing the impact of national ART policy changes in México. Here, we present a four-year observational study describing PDR trends in Mexico City from 2017 to 2020. The study leverages a scientific collaboration between the largest primary care HIV clinic in Mexico City and a reference HIVDR testing laboratory, performing baseline HIV sequencing in all persons receiving an HIV diagnosis locally. Mexico City encompasses 18% of persons on ART in the country, and its epidemic is highly concentrated in men who have sex with men (MSM), with a high rate of linkage to care and ART use compared with other areas in the country [12].

## 2. Results

### 2.1. Characteristics of the Study Participants

Between January 2017 and December 2020, 8128 blood specimens were collected at the Condesa Specialized Clinic, the largest HIV primary care center in Mexico City. From these, 6785 (83.5%) were successfully sequenced. After curation, removal of duplicates (the first sequence of each individual in this case was kept), and sequence quality filtering (see Methods), 6661 unique individuals with protease-reverse transcriptase (PR.RT) as well as integrase (IN) sequence available were included in the database. Taking into account different amounts of missing information for different variables across the data set (Table 1), 95.1% (6234/6555) of the participants were cisgender men, 4.1% (267/6555) cisgender women, 0.8% (51/6555) transgender women, and 0.05% (3/6555) transgender men; 80.0% (4223/5280) lived in Mexico City and 18.3% (967/5280) in the surrounding municipalities of the State of Mexico; and 39.7% (2323/5853) arrived with <200 CD4+ T cells/mm^3^ to clinical care. Subtype B largely predominated in the study population with only 1.3% (84/6661) of participants having non-B subtypes. The most frequent non-B subtypes observed were circulating recombinant forms (CRF02_AG: 0.29%; CRF01_AE: 0.26%) and unique recombinant forms (BG: 0.20%; BF: 0.17%).

Considering a subgroup of 1348 participants enrolled from June to December 2020, for whom more detailed metadata were collected (Table 2), and accounting for missing data, an estimated 11.0% (141/1286) reported previous exposure to ART, 59.9% (717/1196) self-identified as belonging to the middle social class, 40.3% (484/1201) had at least an undergraduate degree, and 3.3% (40/1196) reported speaking any indigenous language. Regarding additional risk variables, 40.2% (426/1059) reported having a sexually transmitted infection in the previous 6 months, 78.1% (793/1016) reported a receptive role in anal sex, 77.1% (807/1047) were not circumcised, and 9.2% (93/1013) reported using venues for sex (Table 2).

### 2.2. Overall Estimations of Pretreatment Drug Resistance

After removal of sequences with quality control issues, 6688 PR-RT and 6709 IN sequences were available for HIVDR analysis (a total of 6661 participants had both PR-RT and IN sequences available). Considering the complete study period, overall PDR to any drug class in Mexico City was 14.4% (95% CI: 13.6–15.3%) (Figure 1a). PDR to NNRTI was higher than to any other drug class (12.0%, 95% CI: 11.3–12.8%, *p* < 0.0001), and exceeded the 10% threshold recommended by the WHO for public health action. PDR to NRTI was 4.2% (3.7–4.7%), and to PI, 3.1% (2.7–3.6%). PDR to any INSTI was the lowest (1.0%, 0.8–1.3%, *p* < 0.0001). In agreement with the preferential use of efavirenz-based first-line regimens in Mexico during most of the analysis period, PDR to efavirenz/nevirapine was 10.4% (9.7–11.2%). Nevertheless, PDR to efavirenz/nevirapine + any NRTI remained low (1.1%, 0.9–1.7%). Importantly, given the current preferential use of INSTI-based first-line regimens, PDR to first-generation INSTI was 1.0% (0.8–1.3%) and to second-generation INSTI 0.3% (0.2–0.4%, *p* < 0.0001).

Considering individual drugs, PDR to nevirapine (10.4%, 9.7–11.6%) and to efavirenz (9.3%, 8.6–10.0%) was the highest across the study period, followed by PDR to rilpivirine (5.1%, 4.6–5.7%) (Figure 1b). Emtricitabine and tenofovir showed the lowest PDR among NRTIs (0.5%, 0.4–0.8%, and 1.1%, 0.9–1.4%, respectively). Among the currently used drugs as third component, PDR to boosted darunavir (0.3%, 0.2–0.5%), dolutegravir (0.2%, 0.1–0.4%) and bictegravir (0.1%, 0.0–0.2%) was low.

The most frequent surveillance drug resistance mutations (SDRMs) were K103NS (6.7%) to NNRTI; M41L (1.2%) and T215CDEF (2.1%) to NRTI; M46IL (1.7%) to PI; and E138AKT (0.2%), Q148HKR (0.1%), and S230R (0.1%) to INSTI (Figure 2a). However, other polymorphic non-SDRMs were also frequent (V106I (3.6%), V179DE (7.1%) in RT) (Figure 2b). Given the low prevalence of non-B subtypes in the study population, associations between specific viral subtypes and the presence of polymorphic mutations were not particularly relevant in the context of the study population, e.g., V106I was present in all F1 subtypes; however, only 6/6661 F1 viruses were observed, while V106IM (generally combined with other NNRTI mutations) was observed in 231/6577 (3.5%) subtype B viruses.

We explored associations of PDR with demographic, clinical, and risk variables collected for a subgroup of participants starting from June to December 2020. As expected, persons with prior exposure to ARV had overall significantly higher prevalence of PDR (*p* < 0.01) (Table 2, Figure 3). This was true for NNRTI and NRTI, but not for boosted PI or INSTI (Figure 3). Persons who inject drugs also showed higher PDR level (*p* = 0.04). Interestingly, persons with incomplete metadata collection in general (who failed to answer several questions of the questionnaire) showed higher PDR (*p* < 0.05). Additionally, persons living in a domestic partnership, with a high school level education, who identified as heterosexual cisgender men, who were circumcised, and who engaged more frequently in insertive sexual relations showed lower PDR to efavirenz/nevirapine (all *p* < 0.05) (Table 1). In general, persons with PDR had slightly lower viral load (*p* < 0.05) (Table 2).

The group of participants with prior exposure to ARVs showed significant differences in demographic, clinical, and risk characteristics in comparison to ART-naïve participants, including a higher frequency of cisgender women (12.8 vs. 5.6%), lower frequency of cisgender MSM (60.3 vs. 68.3%), higher age (median: 35 vs. 29 years), lower education (elementary or none: 12.2 vs. 4.8%), higher frequency of persons who inject drugs (8.5 vs. 3.1%), and lower use of apps for finding sexual partners (31.9 vs. 42.8%) (*p* < 0.05 in all cases; Appendix A). Considering only ART-naïve persons enrolled from June to December 2020, HIVDR reached 10.2% (CI 95%: 8.5–12.1%) to efavirenz/nevirapine, 4.0% (3.0–5.3%) to NRTI, 1.7% (1.1–2.7%) to boosted atazanavir, lopinavir or darunavir, and 1.4% (0.8–2.3%) to any INSTI (Figure 3). Of note, no cases of bictegravir or dolutegravir resistance were observed among prior ARV-exposed participants (only 1 participant (0.7%) showed cabotegravir resistance due to the N155H mutation).

### 2.3. Pretreatment Drug Resistance Trends across the Study Period

A significant increasing trend for NNRTI PDR was observed from 2017 to 2020 (10.3 to 13.6%, *p* = 0.02). PDR to all other drug classes remained stable: NRTI below 5%, PI below 3.5%, and INSTI below 1% (Figure 4a). Importantly, no increase in PDR to second-generation INSTI was observed, remaining under 0.3% across the complete study period.

The observed increase in NNRTI PDR was mainly attributable to increasing PDR to efavirenz (8.0 to 10.4%; *p* = 0.05), but also to rilpivirine (4.5 to 5.9%, *p* = 0.03) (Figure 4b). No other drugs showed significant trends (Appendix A).

In the Mexican context, PDR to all relevant drugs for currently recommended first-line ART regimens remained stable and below 2%, except for efavirenz (Figure 4c). Regarding possible NRTI backbones, PDR to abacavir, emtricitabine/lamivudine, and tenofovir, remained low (under 2%) and stable. The same was true for boosted darunavir (under 0.5%), and dolutegravir/bictegravir (under 0.3%).

A significant increasing trend was observed in the prevalence of any RT SDRM across the study period (10.6 to 12.5%, *p* = 0.04), particularly to any NNRTI SDRM (7.3 to 10.0%, *p* = 0.04). The prevalence of K103NS varied from 5.2 to 7.7%, but this increase was not significant (*p* = 0.08) (Figure 4d). No other DRM (including both SDRM and non-SDRM) showed significant prevalence changes during the period (Appendix A).

### 2.4. Associations of Pretreatment Drug Resistance with Epidemiological Variables

We analyzed possible associations between the presence of PDR and demographic and clinical variables. This analysis encompassed only individuals enrolled from June 2020 to December 2020, a period in which variables were collected through a computer-based questionnaire and the completeness of the metadata was acceptable. After multivariable adjustment, the presence of PDR to any drug remained strongly associated with prior exposure to ARV drugs (adjusted odds ratio, aOR: 2.7, 95% CI: 1.7–4.1, *p* = 0.001). The odds of PDR were higher in persons presenting to care with <200 CD4+ T cells/mm^3^ compared to 200–499 CD4+ T cells/mm^3^ (aOR: 1.6, 95% CI: 1.2–2.2, *p* = 0.005) (Table 3).

### 2.5. Analysis of HIV PDR Transmission within Mexico City’s HIV Genetic Network

We identified clusters of individuals with PDR within Mexico City’s HIV transmission network. The network was inferred from 6688 PR-RT sequences from individuals arriving to care from 2017 to 2020, from which 2960 (44.3%) were found to belong to 820 clusters, ranging in size from 2 to 41 nodes (Figure 5). No difference was observed between clustering and non-clustering individuals in the proportion of viruses with PDR to any drug (13.9 vs. 14.7%, *p* = 0.65) or to efavirenz/nevirapine (9.7 vs. 11.0%, *p* = 0.15). A total of 19.4% (159/820) of all clusters included at least one person with PDR and 14.8% (121/820) included at least one person with resistance to efavirenz/nevirapine. Resistance to efavirenz/nevirapine was associated with transmission of K103NS alone in 7.9% (65/820) of clusters, K103NS plus other NNRTI mutations in 3.4% (28/820) clusters and with other NNRTI mutations in 8.0% (66/820) of clusters (Figure 5). In 41.3% (50/121) of the clusters with efavirenz/nevirapine PDR, resistance was shared by 100% of the nodes, and in 32.4% (38/121) by 50 to 99% of the nodes. Some examples of the larger clusters evidencing NNRTI PDR transmission included cluster NNRTI-1, with 14 nodes, all men living in Mexico City with median age 28 years (IQR 23–31), all of them with K103NS; cluster NNRTI-2, with 27 nodes formed by men with median age 25 (23–30) enrolled across the complete study period, 11% (3/27) with K103NS and 74% (20/27) with other mutations; NNRTI-3, with 21 nodes, including men living in Mexico City and the State of Mexico and median age 24 (20–29), 62% (13/21) with K103NS; and cluster NNRTI-4, with 12 nodes, all men with median age 28 (25–32), all sharing E138A (Figure 5).

A total of 2.7% (22/820) of the clusters included at least one person with INSTI PDR and only 0.7% (6/820) with at least one person with bictegravir/dolutegravir resistance. The largest clusters with INSTI PDR transmission were cluster INSTI-1, with 5 nodes sharing the S230R mutation, formed by 4 cisgender men and 1 transgender woman, with median age 23 (20–32); and cluster INSTI-2, with 4 nodes, all men sharing the E157Q and G163K mutations and median age 28 (26–34).

Other interesting clusters evidencing PDR transmission within the network included cluster PI-1, with 29 nodes, all men with PI resistance and median age 24 (22–28) enrolled across all years; cluster NRTI-1, with 23 nodes, 22 cisgender men and 1 cisgender woman, with median age 22 (10–27) and constant growth across all years; cluster complex-1, with 10 nodes, all men enrolled from 2018 to 2020 and median age 23 (20–30), 60% (6/10) with PI + NRTI + NNRTI resistance and 40% (4/10) with PI + NRTI resistance; and complex-2, with 7 nodes, all men with median age 26 (21–31) sharing PR M46I, L90M, RT M41L, D67N, T69D, L210W, T215D and RT K103N, Y181C (Figure 5).

### 2.6. Characteristics and Prevalence of Acquired Drug Resistance in Mexico City, 2020

In order to describe acquired drug resistance in the context of ART optimization in Mexico City, starting in 2019, we analyzed all clinically indicated HIV genotypes for persons cared for at the Condesa clinics in 2020, who had two consecutive viral load estimations > 1000 copies/mL. A total of 143 individuals were eligible (see Methods). From these, we obtained 142 successful PR-RT sequences and 137 IN sequences. After quality filtering, 133 PR-RT and 128 IN sequences were used for the analysis. Overall, 39.8% (95% CI: 31.5–48.7%) individuals had ADR to any drug class (Figure 6a). Considering individual drug classes, 34.6% (26.6–43.3%) of individuals showed ADR to NNRTI, 26.3% (19.1–34.7%) to NRTI, 1.5% (0.2–5.3%) to boosted PI, and 2.3% (0.5–6.7%) to INSTI. A total of 21.1% (14.5–29.0%) showed resistance to NRTI and efavirenz/nevirapine. Considering individual drugs, 33.8% individuals were resistant to efavirenz, 23.3% to emtricitabine/lamivudine, and 8.3% to tenofovir. Resistance to dolutegravir/bictegravir was 2.3% and to darunavir 0%. Regarding other drugs commonly used in second or third line, ADR to etravirine was 14.3%, 18.8% to rilpivirine, 20.3% to doravirine, and 2.3% to zidovudine (Figure 6b). The most commonly observed DRMs were K103NS (24.1%) and M184VI (23.3%) in RT. Resistance in IN was mainly due to R263K (1.6%) and E138AKT, G140ACS, Q148HKR (0.8% each) (Figure 6c). The relatively high resistance to doravirine is noteworthy, and was mainly associated with mutations Y188L (4.5%, in most cases combined with V106I), L100IV (4.5%), K103EP (3.8%), P225H (3.0%).

## 3. Discussion

The present study showed an increasing trend of PDR to efavirenz in Mexico City from 2017 to 2020, crossing the 10% threshold by the end of the study period, while PDR to dolutegravir and bictegravir remained low and under 0.5%. Given the high genetic barrier of second-generation INSTI-based ART regimens [13], as well as the low impact of baseline DRMs observed in the NRTI backbone in the effectiveness of these regimens [14], HIV PDR surveillance has become less of a priority in regions widely using dolutegravir and bictegravir. Nevertheless, given the historical rapid increase in NNRTI resistance in LMICs worldwide, it is important to maintain surveillance in order to detect increasing PDR trends in time and save important ART options for future management of HIV, strengthening HIV programs with focus on HIVDR-associated early warning indicators. Focused HIVDR surveillance to identify specific regions and populations with specific vulnerability issues and programmatic gaps is also important to improve the HIV care continuum with focused interventions and to optimize both first- and second-line ART regimens, as well as pre- (PrEP) and post-exposure prophylaxis (PEP) regimens.

As expected, PDR to efavirenz/nevirapine continued to grow in Mexico City across the study period, maintaining the previously observed trend [10]. Importantly, no changes in this trend were observed after the nationwide implementation and procurement of second-generation INSTI-based regimens as preferred first-line options since the second half of 2019, which suggests onward transmission of efavirenz resistance even in the context of a significantly reduced use of the drug. Indeed, when analyzing Mexico City’s HIV transmission network for the study period, we observed 88 clusters for which at least 50% of the nodes shared NNRTI PDR mutations. These included large clusters formed mainly by young cisgender men sharing the K103NS mutation and other NNRTI PDR-associated mutations. It is also important to consider the possible impact of ongoing DRM transmission on the growing cross-resistance trend to rilpivirine observed in the present study for the future use of long-acting regimens as part of both first- and second-line ART as well as PrEP regimens.

As expected, efavirenz/nevirapine resistance and, in general, PDR to any ARV drug were strongly associated with prior exposure to ARVs. The group of persons with prior exposure to ARVs in Mexico City was mostly formed by persons that, after being lost to clinical follow-up, later returned to clinical care to restart ART, keeping in mind that PrEP is not yet widely available in Mexico. This group of ART restarters represents an important challenge and target for possible interventions, and it showed specific characteristics that were different from the rest of the study population, including higher representation of cisgender women and transgender women, higher age, lower education in general and higher representation of injectable drug users (Appendix A). Some of these characteristics were described previously in a nationwide PDR study [9]. Unexpectedly, the odds of having PDR were not specifically higher in cisgender women, as observed in previous studies in Mexico and in other countries [5,9]. In the context of Mexico City, contrasted with other regions of Mexico, the epidemic is highly concentrated in MSM [15] who are generally younger, have a higher education level, and arrive earlier to clinical care than the heterosexual population [16,17]. In this context, MSM had significantly higher odds of efavirenz/nevirapine PDR compared to heterosexual cisgender men, possibly reflecting ongoing transmission of NNRTI resistance mutations such as K103NS and E138A. On the other hand, it is reassuring that both PDR and ADR to dolutegravir and bictegravir in 2020 remained low, although further surveillance is warranted given the short time of implementation of ART optimization nationwide. Taking this into consideration, most ADR cases observed in the present study could still be associated with failures to offer efavirenz-based first-line regimens in persons who had not yet switched to INSTI-based options, which is also consistent with the high frequency of K103NS and M184VI observed in persons with ADR. Also noteworthy is the fact that persons arriving to care with advanced infection had higher odds of PDR. This could be associated with ART defaulters who, years later, return to clinical care because of complications associated with opportunistic infections, but could also suggest a subpopulation of MSM characterized by late arrival to clinical care. Interestingly, this study identified a subset of individuals that did not answer several of the questions in the computer-based questionnaire and that were characterized overall by significantly higher PDR (Table 2). Further studies including in-depth interviews could be highly valuable for understanding this group, as it may present common vulnerability issues. Finally, although the level of ADR to dolutegravir/bictegravir observed in 2020 was low, it still warrants strengthening of strategies to improve adherence both at the clinic and community levels, especially in groups with a high risk of ART defaulting. This is especially relevant given that the overall level of ADR observed in this study was lower than that observed in other countries [5], possibly suggesting more frequent interruption of ART in persons without viral suppression. Also important is the high prevalence of cross-resistance to doravirine in persons with ADR. This drug is not yet available in Mexico, but the current level of cross-resistance observed could limit its use locally.

The present study has important limitations worth mentioning. First, all participants were enrolled at a single institution, which may cause selection bias. Even though the Condesa clinic encompasses an important proportion of new diagnoses in the metropolitan area of Mexico City, a fraction of the population living with HIV could be underrepresented, especially persons living in areas with local social security clinics that offer HIV testing within the city. Nevertheless, even though the Condesa clinics care for persons lacking health insurance, 42% of HIV diagnoses performed at the clinics are in persons with social security [15], from which approximately 20% are lost to follow-up and most probably later return to clinical care in subsequent years or die. This fact strengthens the representativeness of the study population, even when coming from a single center. Second, information bias could exist, especially given that metadata collection was poor, especially at the beginning of the study period. Still, given its role as the most important HIV diagnosis center in Mexico City and the size of the clinic, we expect the study population to be highly representative of the population of persons living with HIV locally. Third, the present study excludes important populations that warrant further studies in both their specific vulnerabilities and structural challenges and their contribution to HIV PDR, especially male adolescents who have sex with men.

## 4. Materials and Methods

### 4.1. Study Population

A cross-sectional, observational study was conducted at the Condesa Specialized Clinic, with two branches in Mexico City (located in the municipalities of Cuauhtémoc and Iztapalapa), between 2017–2020. The Condesa Specialized Clinic is the largest primary HIV care clinic and one of the main HIV diagnostic centers in Mexico, having diagnosed nearly 3500 individuals in 2020, approximately 70% of all the new infections in Mexico City’s metropolitan zone [15]. All adults (>18 years) attending the Condesa Clinic for an HIV test, having received a positive result, including new diagnoses, referrals from other institutions, and persons returning to care after at least 3 months of ART defaulting, were invited to participate in the study between January 2017 and December 2020. These inclusion criteria were defined according to the WHO PDR definition [5]. Participants gave written informed consent to participate in the study. Since 2020, a computer-based self-administered questionnaire including demographic and clinical data was applied (a paper-based version of the questionnaire was available for participants preferring this option). Analysis of the variables included in the questionnaire was performed in a subset of the participants enrolled from June 2020 to December 2020. All participants donated a blood specimen for HIV sequencing and DR testing. HIV sequencing was performed at the Center for Research in Infectious Diseases of the National Institute of Respiratory Diseases (CIENI/INER), a reference center for HIV genotyping, following strict quality assurance processes. The study was reviewed and approved by the Institutional Review Board of the National Institute of Respiratory Diseases (project codes E12-17 and E02-20) and was conducted according to the principles of the Declaration of Helsinki.

Analysis of the characteristics and prevalence of acquired drug resistance was performed in an independent study group, including all clinically indicated HIV genotypes recorded at the National Ministry of Health HIV Database (SALVAR) from January 2020 to December 2020, as part of the national HIV program. HIV genotyping is recommended by the Mexican ART Guidelines for switching to second- and third-line regimens. All persons cared for at the Condesa clinic with two consecutive viral load values > 1000 copies/mL, whose second viral load test was performed in 2020, were included in the analysis. HIV sequencing was performed at CIENI/INER from the same blood specimen donated for the second viral load test, using internally validated Sanger sequencing methods, as explained below.

### 4.2. HIV Amplification and Sequencing

Sequences were obtained by next generation sequencing from a single amplicon including the complete protease (PR), reverse transcriptase (RT) and integrase (IN) genes (HXB2: PR 1-99, RT 1-560 and IN 1-288), using an in-house-validated method with Illumina sequencing technology on a MiSeq instrument (San Diego, CA, USA), as previously described [16,17]. A minority of the specimens in which amplification of this longer amplicon was not successful, as well as clinically indicated HIV genotypes for the ADR analysis, were amplified using a validated protocol developed by the US Centers for Disease Control and Prevention for the PR-RT region (HXB2 positions: PR 6-99, RT 1-251) [18] and an in-house developed and validated protocol for IN (HXB2: IN 1-288) [19]. These shorter amplicons were sequenced using NGS with standard Illumina protocols or by Sanger sequencing on a 3730xl Genetic Analyzer (ThermoFisher, Waltham, MA, USA) as previously described [10].

Next generation sequencing reads were filtered and assembled using HyDRA (Public Health Agency of Canada, Winnipeg, MB, Canada) [17,20]. Twenty percent consensus sequences (previously validated as Sanger-like sequences) were obtained and used for the HIVDR analyses [21]. Sanger sequences were assembled and edited using ReCall (BC Centre for Excellence in HIV/AIDS, Vancouver, BC, Canada) [22].

### 4.3. HIV Drug Resistance Assessment

Quality controls were applied to the sequences included in the database using the WHO HIVDR quality control tool [23,24]. Sequences not compliant with quality control were excluded from the study. Reasons for exclusion included inadequate sequence length, presence of stop codons, frameshift insertions/deletions, excess Apolipoprotein B mRNA-Editing Catalytic Polypeptide-like (APOBEC) or unusual mutations [23,24]. For participants with more than one sequence available, the first sequence was selected.

PDR was estimated using the Stanford HIVdb tool V.9.0 [24] and reported by drug class and individual drugs. Sequences with HIVDR were defined as those with a Stanford score ≥ 15 (at least low-level resistance) for efavirenz, nevirapine, any nucleoside reverse transcriptase inhibitor (NRTI), boosted darunavir, lopinavir, or atazanavir, raltegravir, elvitegravir, dolutegravir, or bictegravir, according to WHO standardized protocols [25]. PDR to INSTI was also reported referring to first-generation (raltegravir and elvitegravir) and second-generation INSTI (dolutegravir, bictegravir, and cabotegravir). HIVDR prevalence was estimated using a predefined Excel template developed for the WHO HIV ResNet Laboratory Network [26] (available by request from the corresponding author). HIV subtype was inferred using the REGAHIV-1 Subtyping Tool version 3.0 [27].

### 4.4. Associations between Pretreatment Drug Resistance and Epidemiological Variables

Exploratory analyses were performed comparing persons with and without PDR to any drug or with PDR to efavirenz/nevirapine, using Mann–Whitney U, chi square, or Fisher’s exact tests, in accordance with the type of variable. Age and viral load were analyzed as continuous variables; gender, state of residence, marital status, education, social class, sexual risk, injectable drug use, use of venues for sex, and previous exposure to ARV were analyzed as categorical variables. CD4+ T cell count was stratified according to CDC clinical categories. Univariate associations between demographic, clinical and behavioral variables available and the presence of PDR to any ARV drug or PDR to efavirenz/nevirapine were explored with logistic regression, including only participants enrolled from June to December 2020, when completion of the metadata was best. Multivariable logistic regression models were constructed using all variables significantly associated with PDR in the univariate analyses. Additional variables were included a priori, owing to previous interest in HIVDR development. The best model was selected using Akaike information criterion, Bayesian information criterion, and Hosmer–Lemeshow goodness of fit test. Analyses were performed using STATA v16.

### 4.5. HIV Transmission Network Inference

The network was defined using a genetic matrix method based on PR-RT sequences, with Seguro HIV-TRAnsmission Cluster Engine (Seguro HIV-TRACE) [16], a locally adapted and secured version of the HIV-TRACE tool [28]. Clusters were defined when sequences showed pairwise Tamura–Nei 93 genetic distance < 1.5%. Although IN sequences were not used for network inference, the presence or absence of HIVDR to INSTI was considered as an additional attribute of each node.

## 5. Conclusions

The present study showed a continued increasing trend of NNRTI PDR in the context of Mexico City’s HIV epidemic in 2017–2020, which was strongly associated with ongoing NNRTI DRM transmission, even with a much lower efavirenz use. It is reassuring that both PDR and ADR to dolutegravir and bictegravir have remained low locally after the widespread rollout of ART first-line regimens based on these drugs locally. Our study also identified demographic characteristics of groups with higher risk of PDR and higher probability of becoming lost to follow-up, in particular, ART restarters. The observations of the present study may be useful to guide clinical care algorithm design within the Condesa clinics in order to identify persons with these profiles who may need specific interventions differentiated from the rest of the population. In the same sense, our observations warrant further in-depth studies in the population of cisgender women, young MSM, and adolescents in order to design better strategies for retention in care and viral suppression.

## Figures and Tables

**Figure 1 pathogens-10-01587-f001:**
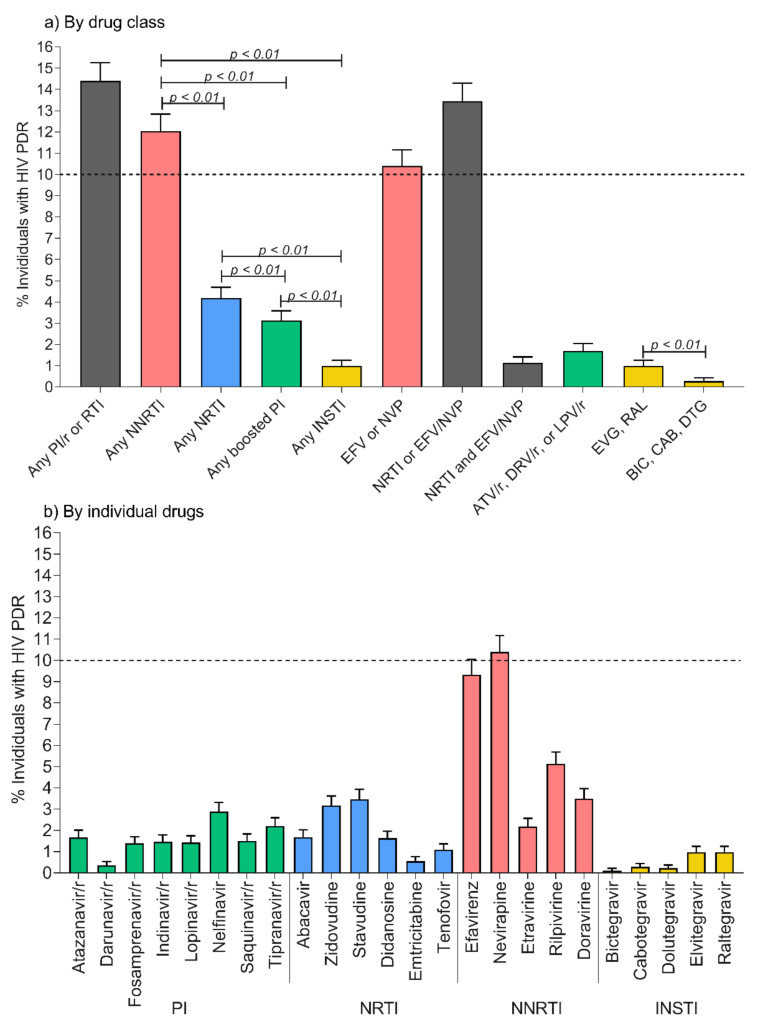
Overall pretreatment drug resistance levels in Mexico City, 2017–2020. Classified by (**a**) drug class, (**b**) individual drug. PDR was estimated from 6688 protease-reverse transcriptase and 6709 integrase sequences using the Stanford HIVdb tool (v9.0). Individuals with HIV PDR were defined as those having a score ≥15 to any of the drugs in the corresponding category, as defined in the Methods. Error bars represent 95% confidence intervals. Comparisons were made with Fisher’s exact test. PDR, pretreatment drug resistance; PI, protease inhibitor; RTI, reverse transcriptase inhibitor; NRTI, nucleoside reverse transcriptase inhibitor; NNRTI, non-nucleoside reverse transcriptase inhibitor; INSTI, integrase strand-transfer inhibitor; EFV, efavirenz; NVP, nevirapine; ATV/r, atazanavir/ritonavir; DRV/r, darunavir/ritonavir; LPV/r, lopinavir/ritonavir; EVG, elvitegravir; RAL, raltegravir; BIC, bictegravir; CAB, cabotegravir; DTG, dolutegravir.

**Figure 2 pathogens-10-01587-f002:**
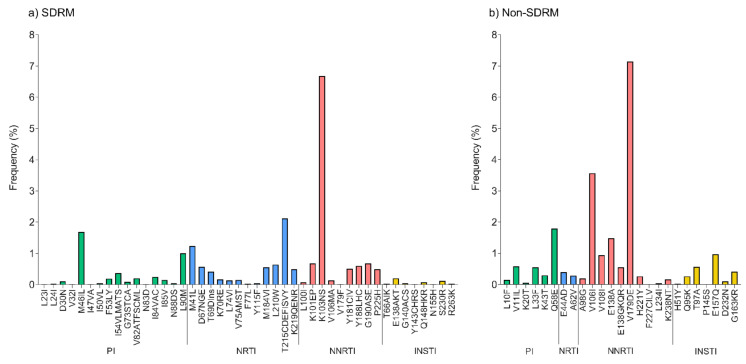
Drug resistance mutation frequency in Mexico City, 2017–2020. Frequencies of surveillance drug resistance mutations (SDRMs) (**a**) and non-SDRMs (**b**) for the complete study period are shown, including a total of 6688 protease-reverse transcriptase and 6709 integrase sequences from unique individuals. Only resistance-associated mutations observed in the sequences are shown. All SDRMs and non-SDRMs defined in the Stanford HIV drug resistance database were analyzed. PI, protease inhibitors; NRTI, nucleoside reverse transcriptase inhibitors; NNRTI, non-nucleoside reverse transcriptase inhibitors; INSTI, integrase strand-transfer inhibitors.

**Figure 3 pathogens-10-01587-f003:**
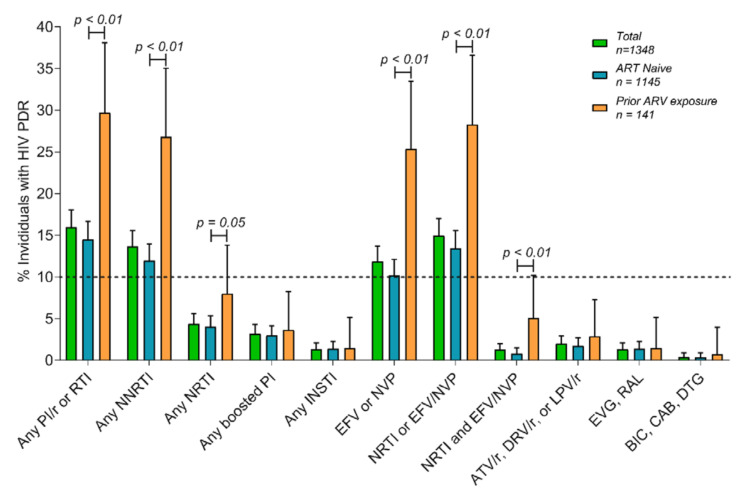
Pretreatment drug resistance in ART-naïve and prior ARV-exposed individuals in Mexico City, 2020. PDR levels are shown for a subset of 1348 participants enrolled from June to December 2020 (data on ARV exposure for 62 participants are missing), for whom data on prior exposure to antiretrovirals were available. Individuals with HIV PDR were defined as those having a score ≥ 15 to any of the drugs in the corresponding category, as defined in the Methods. Error bars represent 95% confidence intervals. Comparisons were made with Fisher’s exact test. PDR; pretreatment drug resistance; PI, protease inhibitor; RTI, reverse transcriptase inhibitor; NRTI, nucleoside reverse transcriptase inhibitor; NNRTI, non-nucleoside reverse transcriptase inhibitor; INSTI, integrase strand-transfer inhibitor; EVG, elvitegravir; RAL, raltegravir; BIC, bictegravir; CAB, cabotegravir; DTG, dolutegravir.

**Figure 4 pathogens-10-01587-f004:**
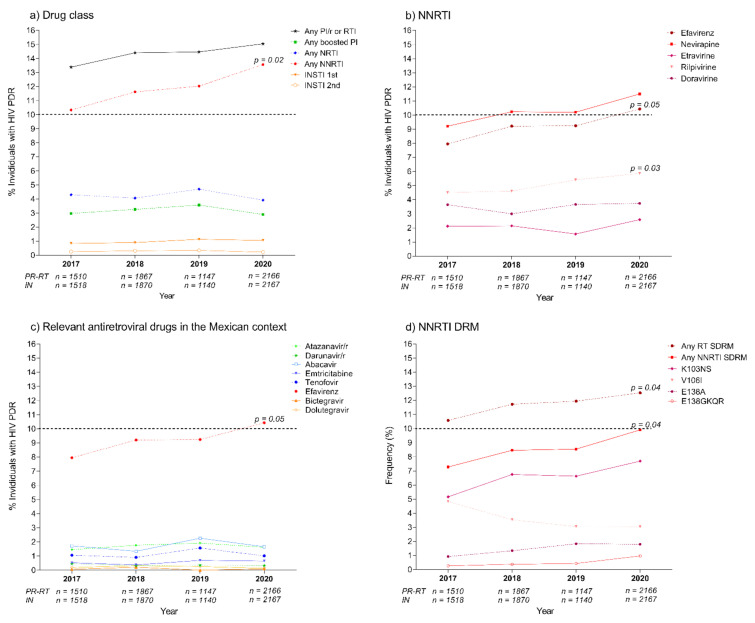
Pretreatment drug resistance trends in Mexico City, 2017–2020. (**a**) By drug class, (**b**) For NNRTI, (**c**) Including relevant antiretroviral drugs in the Mexican context, and (**d**) Including selected drug resistance mutations with relevance to NNRTI PDR. PDR was estimated from protease (PR), reverse transcriptase (RT) and integrase (IN) sequences using the Stanford HIVdb tool (v9.0). Individuals with HIV PDR were defined as those having a Stanford score ≥ 15 for any of the drugs in the corresponding category, as described in the Methods. DRMs included SDRM and non-SDRM. First-generation INSTIs include raltegravir and elvitegravir; second-generation INSTIs include dolutegravir, bictegravir and cabotegravir. Trends were tested using linear regression. PDR, pretreatment drug resistance; PI, protease inhibitors; RTI, reverse transcriptase inhibitors; NRTI, nucleoside reverse transcriptase inhibitors; NNRTI, non-nucleoside reverse transcriptase inhibitors; INSTI, integrase strand-transfer inhibitors; DRM, drug resistance mutation; SDRM, surveillance DRM.

**Figure 5 pathogens-10-01587-f005:**
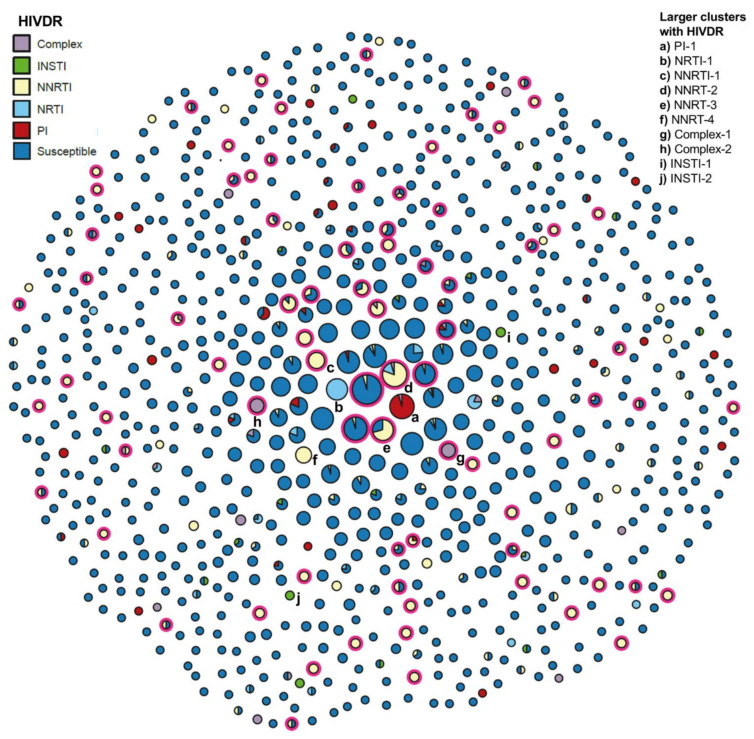
PDR transmission within Mexico City’s HIV genetic network, 2017–2020. The network was inferred from 6688 protease-reverse transcriptase sequences from individuals arriving to care from 2017 to 2020, using a locally adapted version of the HIV-TRACE tool. Each circle represents a cluster. The size of the circle reflects the size of the cluster. Clusters are colored according to the presence of pretreatment drug resistance by drug class. Specific clusters mentioned in the text are identified with letters on the lower-right side. Red circles surrounding clusters show the presence of viruses with the K103NS mutation.

**Figure 6 pathogens-10-01587-f006:**
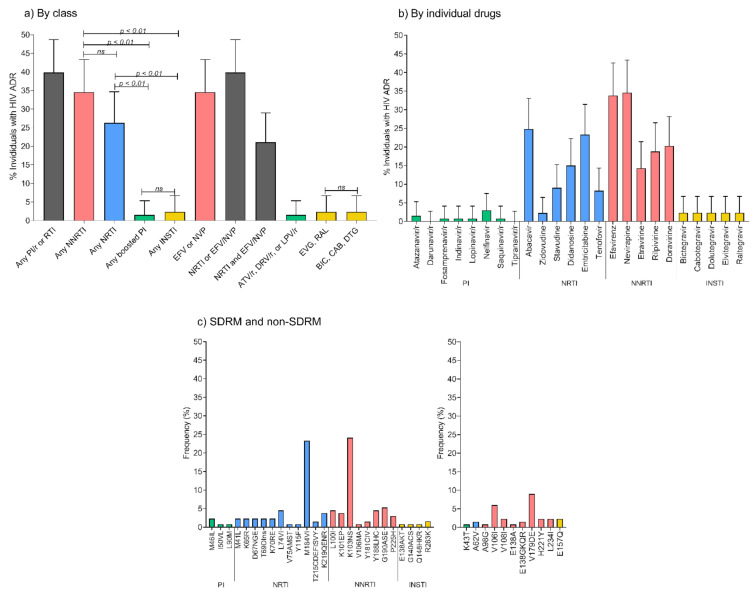
Acquired drug resistance in Mexico City, 2020. (**a**) By drug class, (**b**) by individual drugs, (**c**) SDRM and non-SDRM. ADR was estimated from protease (PR), reverse transcriptase (RT) and integrase (IN) sequences using the Stanford HIVdb tool (v9.0). Individuals with HIV ADR were defined as those having a Stanford score ≥ 15 for any of the drugs in the corresponding category, as described in the Methods. ADR, acquired drug resistance; PI, protease inhibitors; RTI, reverse transcriptase inhibitors; NRTI, nucleoside reverse transcriptase inhibitors; NNRTI, non-nucleoside reverse transcriptase inhibitors; INSTI, integrase strand-transfer inhibitors; EFV, efavirenz; NVP, nevirapine; ATV/r, atazanavir/ritonavir; DRV/r, darunavir/ritonavir; LPV/r, lopinavir/ritonavir; EVG, elvitegravir; RAL, raltegravir; BIC, bictegravir; CAB, cabotegravir; DTG, dolutegravir; DRM, drug resistance mutation; SDRM, surveillance DRM.

**Table 1 pathogens-10-01587-t001:** Baseline demographic and clinical characteristics of study participants, 2017–2020 ^a^.

	Total Number of Observations ^b^ *n* = 6661	With PDR to Any Drug ^c,d^ *n* = 957	*p* Value	With PDR to EFV/NVP ^d^ *n* = 690	*p* Value
Gender, *n* (%) Cisgender Women Cisgender Men Transgender Women Transgender Men Missing	267 (4.0) 6234 (93.6) 51 (0.8) 3 (0.0) 106 (1.6)	32 (12.0) 907 (14.5) 6 (11.8) 0 (0.0) 12 (11.3)	0.14 0.06 0.39 0.62 0.23	24 (9.0) 653 (10.5) 4 (7.8) 0 (0.0) 9 (8.5)	0.26 0.13 0.38 0.72 0.33
Age (years) ^e^ Median (IQR)	28 (24–35)	28 (24–34)	0.09	28 (24–34)	0.30
State of residence, *n* (%) Mexico City Mexico State Other Missing	4223 (63.4) 967 (14.5) 90 (1.4) 1381 (20.7)	621 (14.7) 142 (14.7) 9 (10.0) 185 (13.4)	0.16 0.40 0.15 0.13	448 (10.6) 107 (11.1) 5 (5.6) 130 (9.4)	0.20 0.23 0.08 0.10
HIV subtype, *n* (%) B Non-B	6577 (98.7) 84 (1.3)	946 (14.4) 11 (13.1)	0.44	680 (11.6) 10 (11.9)	0.37
Viral load (log RNA copies/mL) ^f^ Median (range)	4.8 (4.25–5–31)	4.7 (4.2–5.3)	0.02 *	4.7 (4.2–5.3)	0.03 *
CD4+ T cell count (cells/mm^3^), *n* ^g^ Median (IQR)	249 (120–395)	250 (123–407)	0.22	254 (129–405)	0.16
CD4+ T cell count category, *n* (%) <200 cells/mm^3^ 200–500 cells/mm^3^ >500 cells/mm^3^ Missing	2323 (34.87) 2721 (40.8) 809 (12.2) 808 (12.1)	334 (14.4) 389 (14.3) 129 (15.9) 105 (13.0)	0.51 0.46 0.10 0.13	240 (10.3) 285 (10.5) 94 (11.6) 71 (8.8)	0.50 0.41 0.12 0.06
CD4+ T cell %, *n* ^h^ Median (IQR)	15 (9–21)	15 (9–22)	0.27	15 (10–22)	0.14

^a^ Data for 6661 participants enrolled from January 2017 to December 2020. ^b^ Column percentages are shown. ^c^ Using the WHO definition for PDR (see Methods). ^d^ Row percentages are shown. ^e^ Data missing for 114 participants (*n* = 6547). ^f^ Data missing for 557 participants (*n* = 6104). ^g^ Data missing for 808 participants (*n* = 5853). ^h^ Data missing for 2234 participants (*n* = 4427). * *p* < 0.05. PDR; pretreatment drug resistance; EFV, efavirenz; NVP, nevirapine; IQR, interquartile range.

**Table 2 pathogens-10-01587-t002:** Baseline demographic, clinical, and risk characteristics of study participants, June–December 2020 ^a^.

	Total Number of Observations ^b^ *n* = 1348	With PDR to Any Drug ^c,d^ *n* = 215 (15.9%)	*p* Value	With PDR to EFV/NVP ^d^ *n* = 158 (11.7%)	*p* Value
Gender, *n* (%) Cisgender Women Cisgender Men Transgender Women Transgender Men	85 (6.3) 1234 (91.5) 28 (2.1) 1 (0.1)	13 (15.3) 196 (15.9) 6 (21.4) 0 (0.0)	0.50 0.46 0.28 0.84	9 (10.6) 145 (11.8) 4 (14.3) 0 (0.0)	0.45 0.53 0.42 0.88
Age (years) Median (IQR)	29 (25–36)	29 (25–35)	0.54	30 (25–35)	0.81
Prior ARV exposure, *n* (%) No Yes Missing	1145 (84.9) 141 (10.5) 62 (4.6)	164 (14.3) 41 (29.1) 10 (16.4)	<0.01 * <0.01 * 0.88	115 (10.0) 35 (24.8) 8 (13.1)	<0.01 * <0.01 * 0.72
State of residence, *n* (%) Mexico City Mexico State Other	1035 (76.8) 279 (20.7) 33 (2.5)	180 (17.4) 32 (11.5) 3 (9.1)	<0.01 * 0.01 * 0.20	134 (13.0) 23 (8.2) 1 (3.0)	<0.01 * 0.02 * 0.08
HIV subtype, *n* (%) B Non-B	1330 (98.7) 18 (1.3)	211 (15.8) 4 (22.1)	0.51	155 (11.6) 3 (16.8)	0.35
Viral load (log RNA copies/mL) Median (range)	4.6 (4.1–5.2)	4.6 (4.0–5.1)	0.22	4.6 (3.9–5.1)	0.20
CD4+ T cell count (cells/mm^3^) Median (IQR)	173 (78–291)	154 (71–281)	0.15	152 (84–284)	0.38
CD4+ T cell count category, *n* (%) <200 cells/mm^3^ 200–500 cells/mm^3^ >500 cells/mm^3^ Missing	757 (56.2) 529 (39.2) 60 (4.4) 2 (0.2)	137 (18.1) 70 (13.2) 8 (13.3) 0 (0.0)	<0.01 * 0.02 * 0.36 0.70	102 (13.5) 51 (9.6) 5 (8.3) 0 (0.0)	0.01 0.03 * 0.28 0.78
CD4+ T cell % Median (IQR)	13 (7–19)	12 (7–18)	041	14 (8–19)	0.48
Sexual risk category, *n* (%) Heterosexual cis women Heterosexual cis men Cisgender MSM Transgender women Missing	78 (5.8) 172 (12.8) 888 (65.9) 31 (2.3) 179 (13.3)	11 (14.1) 20 (11.6) 139 (15.6) 6 (19.3) 39 (21.8)	0.39 0.06 0.36 0.37 0.02 *	8 (10.3) 12 (7.0) 105 (11.8) 3 (9.7) 30 (16.8)	0.42 0.02 * 0.47 0.50 0.02 *
Marital status, *n* (%) Single Married Domestic partnership Other Missing	958 (71.1) 46 (3.4) 173 (12.8) 29 (2.2) 142 (10.5)	153 (16.0) 7 (15.2) 20 (11.6) 3 (10.3) 32 (22.5)	0.53 0.54 0.05 0.29 0.02 *	115 (12.0) 3 (6.5) 13 (7.5) 2 (6.9.6) 25 (17.6)	0.34 0.19 0.04 * 0.32 0.02 *
Education, *n* (%) Illiterate Elementary High School Technician Degree Postgraduate Missing	7 (0.5) 59 (4.4) 587 (43.5) 64 (4.7) 440 (32.6) 44 (3.26) 147 (10.91)	1 (14.3) 11 (18.6) 86 (14.6) 10 (15.6) 69 (15.7) 5 (11.4) 33 (22.4)	0.69 0.33 0.14 0.55 0.46 0.27 0.02 *	1 (14.3) 8 (13.6) 56 (9.5) 9 (14.1) 54 (12.3) 4 (9.1) 26 (17.7)	0.58 0.38 0.02 * 0.33 0.36 0.40 0.02 *
Self-identified social class, *n* (%) Low Middle High Missing	477 (35.4) 717 (53.2) 2 (0.2) 152 (11.3)	74 (15.5) 108 (15.1) 1 (50.0) 32 (21.0)	0.40 0.19 0.29 0.05	50 (10.5) 83 (11.6) 0 (0.0) 25 (16.4)	0.17 0.46 0.77 0.04 *
Indigenous languages spoken, *n* (%) No Yes Missing	1156 (85.7) 40 (3.0) 152 (11.3)	177 (15.3) 6 (15.0) 32 (21.0)	0.07 0.54 0.05	130 (11.2) 3 (7.5) 25 (16.4)	0.11 0.29 0.04 *
Other sexually transmitted infections, *n* (%) No Yes Preferred not to answer Missing	633 (47.0) 426 (31.6) 58 (4.3) 231 (17.1)	91 (14.4) 72 (16.9) 10 (17.2) 42 (18.2)	0.08 0.28 0.45 0.17	72 (11.4) 48 (11.3) 5 (8.6) 33 (14.3)	0.38 0.40 0.31 0.11
Role in anal sex, *n* (%) Receptive Insertive/Receptive Insertive Missing	319 (23.6) 474 (35.3) 223 (16.5) 332 (24.6)	49 (15.4) 76 (16.0) 32 (14.4) 58 (17.5)	0.41 0.50 0.27 0.22	35 (11.0) 64 (13.5) 17 (7.7) 42 (12.6)	0.36 0.08 0.02 * 0.30
Circumcision, *n* (%) No Yes Missing	807 (59.9) 240 (17.8) 301 (22.3)	131 (16.2) 28 (11.7) 56 (18.6)	0.39 0.03 * 0.09	97 (12.0) 20 (8.3) 41 (13.6)	0.37 0.04 * 0.14
Injectable drug use, *n* (%) No Yes Missing	1143 (84.8) 49 (3.6) 156 (11.6)	170 (14.9) 13 (26.5) 32 (20.5)	<0.01 * 0.04 * 0.06	123 (10.8) 10 (20.4) 25 (16.0)	<0.01 * 0.05 0.05
Venues for sex, *n* (%) At home At partner’s home Venues for sex Multiple Missing	519 (38.5) 290 (21.5) 93 (6.9) 111 (8.2) 335 (24.8)	82 (15.8) 48 (16.5) 14 (15.0) 12 (10.8) 59 (17.6)	0.48 0.40 0.47 0.07 0.19	62 (11.9) 36 (12.4) 8 (8.6) 8 (7.2) 44 (13.1)	0.45 0.37 0.22 0.07 0.20

^a^ Data for 1348 participants enrolled from June to December 2020. ^b^ Column percentages are shown. ^c^ Using the WHO definition for PDR (see Methods). ^d^ Row percentages are shown. * *p* < 0.05. ARV, antiretroviral; EFV, efavirenz; NVP, nevirapine; IQR, interquartile range; MSM, men who have sex with men.

**Table 3 pathogens-10-01587-t003:** Risk factors for HIV pretreatment drug resistance in Mexico City, June–December 2020.

	Resistance to Any ARV ^a^			Resistance to NNRTI ^a^		
	aOR	95% CI	*p* Value	aOR	95% CI	*p* Value
Age (years) ^b^	1.0	1.0–1.0	0.07	1.0	1.0–1.0	0.1
Sexual risk category ^c^ Heterosexual cis men Heterosexual cis women Cisgender MSM Transgender women Missing	Ref. 1.1 1.5 1.9 2.8	0.5–2.5 0.9–2.4 0.7–5.4 1.1–7.0	0.8 0.2 0.2 0.03 *	Ref. 1.3 1.9 1.5 3.2	0.5–3.5 1.0–3.5 0.4–5.7 1.1–9.3	0.6 0.1 0.6 0.03 *
Injectable drug use No Yes Missing	Ref. 1.7 0.9	0.9–3.7 0.4–2.0	0.07 0.7	Ref. 1.8 1.0	0.8–3.8 0.4–2.6	0.1 1.0
CD4+ T cell count category 200–499 cells/mm^3^ <200 cells/mm^3^ ≥500 cells/mm^3^	Ref. 1.6 1.0	1.2–2.2 0.4–2.1	0.005 * 0.9	Ref. 1.6 0.8	1.1–2.4 0.3–2.1	0.01 * 0.6
Viral load (log copies/mL) ^b^	0.9	0.7–1.1	0.2	0.9	0.7–1.1	0.2
Prior ARV exposure No Yes Missing	Ref. 2.7 0.9	1.7–4.1 0.4–1.8	0.001 * 0.7	Ref. 3.4 1.0	2.1–5.3 0.4–2.2	0.001 * 0.9

^a^*n* = 1348; ^b^ Analyzed as a continuous variable; ^c^ Sexual risk category was assessed as a composite variable including sex at birth, gender identity and sexual practices. aOR, adjusted odds ratio; CI, confidence interval; ARV, antiretroviral; NNRTI, non-nucleoside reverse transcriptase inhibitors; MSM, men who have sex with men; Ref., reference category. * Statistically significant.

## Data Availability

Data sets supporting the observations of the present study are available upon request from the corresponding author.

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
