# Peer review of "HIV Pretreatment Drug Resistance Trends in Mexico City, 2017–2020"

_pathogens, 2021, doi:10.3390/pathogens10121587_

Round 1

Reviewer 1 Report

Thank the editor for giving me the opportunity to review this important work on HIV pre-treatment drug resistance in monocentric cohort of Mexico City in recent year. The authors show an increasingly high PDR to NNRTI above the threshold of 10% (particularly for EFV and NVP) and low INSTI PDR.  

Even after the transition to 2nd generation INSTI the impact of these findings are relevant given the important role in the future ARV therapies of new NNRTIs (DOR) or new formulation of old NNRTIs (RPV coupled with INSTI regimen in dual regimens (also as LA).

Moreover, the molecular epidemiology approach of this work (especially in the cluster analysis and in the different subgroup analysis) give an important epidemiological view of the HIV transmission and HIV care in the largest city of central-north America. The discussion is very well written.

There are, however, some major flaws, given in most of the cases by the missing demographical and clinical data.

Major comments             

Abstract

  1. Some results described in the abstract, and also cited in the conclusions are only on supplementary materials and never in the results of the manuscript (ART-re starter): those results are important, please add 1 paragraph (or at least 1 sentence) in the results section of the manuscript.

Results

  1. It is really really difficult to follow the numbers of patients included in every step, plese consider to split the Table1 in two:
  • All patients from 2017 (n=6661)
  • Patients from June 2020 (n=1348) with all the related new variables

And change the description of results, accordingly.

Line 80: “working database of 6763 unique individuals” but they become 6661 in the results, why?. 

There are 1162 patients in the logistic regression (2.4 results line 186) vs 1348 (Table 1 Results and Table s1 supplementary). Someone has been excluded in the model for missing data? In case, missing data could be used as a separate group (for each variable where there are missing data, 'missing indicator method').

  1. I believe that it is important to distinguish between ART-Naive and ART-Exposed not in ART among the whole cohort of 6661 patients.

From the epidemiological point of view, those group are different, the ART-exposed group are patients with higher probability of being lost to care other times, and transmitting the virus with its related DR. It would be interesting to see the difference in NNRTI PTDR in naïve and experienced groups in this complete dataset (just as absolute numbers and percentage divided by the two groups).

If it is possible, please repeat the results of 2.2 paragraph with the stratification of naïve and experienced.

If the stratification on the 6661 is not available, you could repeat 2.2 paragraph only in the subset of 1348 subjects of jun-dec20 overall and stratified by naive and experienced (but with a significant loss of representativeness).

  1. Paragraph 2.4: Both in beginning of the paragraph and in the methods below it is stated that cluster network are based on PR/RT sequence, but in the figure and text, you show also clusters with INSTI resistance, please correct.

  2. The ADR group of paragraph 2.5 seems to be a totally different cohort compared to the larger dataset of 6661 or the1348. Why do you use this group, and not the ART-defaulting of previously groups? Do you know the exposure of pervious drugs for each patient? Moreover, is a very small sample size

Methods

  1. Line 353: Cannot find in WHO guidelines [5] the confirmation about the 3 months ARV interruption period, are there some other guideline or literature? “Referrals from other institution”, are also these patients without any ART from at least 3 months?

Minor comment:

  1. Figure 3 and Figure 1S and 2S have some graph duplicates
  2. Figure 5 is in low quality and some of the 95%CI bars are grey
  3. Line 285-286 cluster growth is not shown in results
  4. Cannot find any citation of supplementary materials in the results section, please add some small sentences about that also in the results section.
  5. Some comments in the discussion, on the significantly not low prevalence of DOR resistance in the ADR group (DOR that is also not influenced by the highly prevalent K103N)

Author Response

Reviewer 1

Comments and Suggestions for Authors

Thank the editor for giving me the opportunity to review this important work on HIV pre-treatment drug resistance in monocentric cohort of Mexico City in recent year. The authors show an increasingly high PDR to NNRTI above the threshold of 10% (particularly for EFV and NVP) and low INSTI PDR.  

Even after the transition to 2nd generation INSTI the impact of these findings are relevant given the important role in the future ARV therapies of new NNRTIs (DOR) or new formulation of old NNRTIs (RPV coupled with INSTI regimen in dual regimens (also as LA).

Moreover, the molecular epidemiology approach of this work (especially in the cluster analysis and in the different subgroup analysis) give an important epidemiological view of the HIV transmission and HIV care in the largest city of central-north America. The discussion is very well written.

There are, however, some major flaws, given in most of the cases by the missing demographical and clinical data.

Major comments             

Abstract

  1. Some results described in the abstract, and also cited in the conclusions are only on supplementary materials and never in the results of the manuscript (ART-re starter): those results are important, please add 1 paragraph (or at least 1 sentence) in the results section of the manuscript.

A: We thank the Reviewer for this important comment. We have updated the results to include all the observations of the supplementary materials, particularly those in supplementary table 1, showing demographic differences between ART-naïve and prior ARV-exposed participants: “Interestingly, the group of participants with prior exposure to ARVs showed significant differences in demographic, clinical and risk characteristics in comparison to ART-naïve participants, including a higher frequency of cisgender women (12.8% vs. 5.6%), lower frequency of cisgender MSM (64.4% vs. 77.2%), higher age (median: 35 vs. 29 years), lower education (elementary or none: 12.2% vs. 4.8%), higher frequency of injectable drug users (8.8% vs. 3.5%), and lower use of apps for finding sexual partners (42.9% vs. 54.3%) (p<0.05 in all cases; Supplementary Table 1)” (Results, Section 2.2).

Results

  1. It is really really difficult to follow the numbers of patients included in every step, plese consider to split the Table1 in two:
  • All patients from 2017 (n=6661)
  • Patients from June 2020 (n=1348) with all the related new variables

And change the description of results, accordingly.

A: We apologize for this and thank the Reviewer for the excellent suggestion. We have now divided Table 1 into two: a table containing variables collected since 2017 for all participants and a second one including only participants enrolled from June to December 2020 (n=1348), for whom additional variables were collected. The text has been updated accordingly. It is important to mention that even with this division, an important amount of missing data still exists. This is now accounted for in the analyses and is noted as an important limitation of the study.

Line 80: “working database of 6763 unique individuals” but they become 6661 in the results, why?. 

A: We apologize for this. The final working database was further reduced after curating and performing quality control on the entire set of sequences, leaving 6661 individuals with both PR-RT and IN sequences. These are the ones included in the analyses of the whole database. We have corrected the text to better explain this: “Between January 2017 and December 2020, 8128 blood specimens were collected at the Condesa Specialized Clinic, the largest HIV primary care center in Mexico City. From these, 6785 (83.5%) were successfully sequenced. After curation, removal of duplicates (the first sequence of each individual in this case was kept), and sequence quality filtering (see Methods), 6661 unique individuals with protease-reverse transcriptase (PR.RT) as well as integrase (IN) sequence available were included in the database.” (Results, Section 2.1).

There are 1162 patients in the logistic regression (2.4 results line 186) vs 1348 (Table 1 Results and Table s1 supplementary). Someone has been excluded in the model for missing data? In case, missing data could be used as a separate group (for each variable where there are missing data, 'missing indicator method').

A: As the Reviewer implies, the total number of observations in the model was reduced due to missing data in the different variables included. As suggested, we have re-run the model including missing data as additional strata for each variable, and updated the text to reflect any changes in OR estimations. Overall, results were the same.

  1. I believe that it is important to distinguish between ART-Naive and ART-Exposed not in ART among the whole cohort of 6661 patients.

From the epidemiological point of view, those group are different, the ART-exposed group are patients with higher probability of being lost to care other times, and transmitting the virus with its related DR. It would be interesting to see the difference in NNRTI PTDR in naïve and experienced groups in this complete dataset (just as absolute numbers and percentage divided by the two groups).

If it is possible, please repeat the results of 2.2 paragraph with the stratification of naïve and experienced.

If the stratification on the 6661 is not available, you could repeat 2.2 paragraph only in the subset of 1348 subjects of jun-dec20 overall and stratified by naive and experienced (but with a significant loss of representativeness).

A: We understand that ART-naïve and ART-exposed are different epidemiological groups, but think that it is important to analyze them together using the operational definition of “pretreatment drug resistance”, that the WHO has recommended (https://www.paho.org/en/documents/concept-note-surveillance-hiv-drug-resistance-adults-initiating-antiretroviral-therapy). Inclusion of ART restarters in the surveys is important as they can constitute a significant percentage of persons starting first-line ART in some regions of the world. Also, in order for our study to be comparable with others, we believe that including this group is extremely important.

As the Reviewer suggested, we have included a separate analysis of HIVDR levels in ART-naïve and prior ART-exposed persons, using the better described subcohort of participants enrolled from June to December of 2020, for whom the data is available. Stratification of the complete cohort is not possible due to lack of information on the variable of previous ART exposure, especially for participants enrolled early in the study. We have included a new figure contrasting HIVDR levels by drug class in the two groups. We have also updated the text in section 2.2 of the results.

  1. Paragraph 2.4: Both in beginning of the paragraph and in the methods below it is stated that cluster network are based on PR/RT sequence, but in the figure and text, you show also clusters with INSTI resistance, please correct. 

A: As the Reviewer notes, inference of the HIV network was based on PR-RT sequences only. However, resistance to integrase inhibitors is also available for the participants and it was considered as an additional attribute of each participant. Thus, we can observe clusters of related sequences (inferred with PR-RT sequences) that are also characterized by having resistance to integrase inhibitors. We have clarified this in the methods (Section 4.5).

  1. The ADR group of paragraph 2.5 seems to be a totally different cohort compared to the larger dataset of 6661 or the1348. Why do you use this group, and not the ART-defaulting of previously groups? Do you know the exposure of pervious drugs for each patient? Moreover, is a very small sample size

A: As the Reviewer correctly notes, the ADR group is completely independent and different from the PDR group. It is also different to the ART-restarters included in PDR surveillance. Persons included in the ADR group are persons currently on ART that fail to maintain undetectable viral load for two consecutive measurements, even after adherence counseling. We included all the persons fulfilling this inclusion criteria enrolled at the Condesa clinic in 2020 in this section of the study and did not select for a sample. ART-restarters or defaulters have stopped ART for at least three months and come back to clinical care to restart, without any previous evidence of treatment failure and, in the case of Mexico, are highly enriched in persons lost-to-follow up that are later linked again to clinical care in a more advanced stage of HIV disease or that stop ART due to a change in social security status. In both cases, we have followed recommendations by the WHO to perform HIVDR surveillance, using the recommended definitions. We have tried to clarify this in the methods. Very few data on ADR have been published for Mexico and we believe that including this additional group makes or paper stronger, as it could point to possible future trends in PDR if resistance to integrase inhibitors is high among persons currently being identified with ADR. HIV genotyping for persons failing ART is recommended by national ART guidelines, thus, being a reference center for HIV viral load testing and genotyping, our lab has access to HIVDR data in this group. This data is routinely generated as part of the standard-of-care in Mexico. This is not the case for baseline HIV genotyping, which is not standard of care in the county and needs to be perform as an additional research study.

Methods

  1. Line 353: Cannot find in WHO guidelines [5] the confirmation about the 3 months ARV interruption period, are there some other guideline or literature? “Referrals from other institution”, are also these patients without any ART from at least 3 months?

A: We apologize for failing to include a more appropriate reference. We have now included a reference to the WHO guidelines for performing PDR surveys where the group of prior ARV-exposed is defined. All participants classified in this group fulfill the criteria of having stopped ART for at least three months prior to presentation to clinical care. As we have explained above, in the case of Mexico, this group is highly enriched in persons lost-to-follow up that are later linked again to clinical care in a more advanced stage of HIV disease or that stop ART due to a change in social security status. The health system in Mexico is fragmented and persons with social security are not eligible to receive ART in Ministry of Health clinics and vice versa. This is a frequent reason for treatment interruption and migration between institutions.

Minor comment:

  1. Figure 3 and Figure 1S and 2S have some graph duplicates

A: We thank the Reviewer for this comment. We are aware that some panels repeat information in the supplementary materials. However, we chose to display the data this way for clarity and comparability reasons. We ask the Reviewer to kindly consider the possibility of keeping the presentation of the data as it is in the submitted version.

  1. Figure 5 is in low quality and some of the 95%CI bars are grey

A: We have provided a higher resolution version of the figure.

  1. Line 285-286 cluster growth is not shown in results

A: The comment has been removed from the Discussion.

  1. Cannot find any citation of supplementary materials in the results section, please add some small sentences about that also in the results section.

A: Supplementary Figures 1 and 2 were previously cited in the previously submitted version (results, section 2.3). We have added a paragraph in section 2.2 to describe results shown in supplementary Table 1.

  1. Some comments in the discussion, on the significantly not low prevalence of DOR resistance in the ADR group (DOR that is also not influenced by the highly prevalent K103N)

A: Following the Reviewer’s suggestion, we have added a comment on HIVDR to DOR in section 2.5 of the results and in the discussion.

Reviewer 2 Report

In this study the authors present the results of the first large-scale study of HIV pre-treatment drug resistance at the largest clinic in Mexico. The study included 8128 specimens, within 4 years 6688 genotypes were obtained by full genome sequencing and additionally by Sanger sequencing.

PDR to any drug class was 14.4%, with the largest proportion of mutations formed towards NNRTIs. A very detailed and thorough analysis of the data has been carried out in order to identify the groups of the population most at risk of being infected with drug-resistant strains of the virus, and ways to prevent the transmission of resistant viruses have been outlined.

In addition to traditional indicators, social aspects were studied - education, marital status, behavioral features, etc., the relationship between these parameters and the level of PDR was reliably analyzed.

To be true, I have not seen such a careful analysis of the results on this topic before. This data and method of analysis will be very useful for countries, where baseline HIVDR testing is not standard of care according to national guidelines.

There are no special comments on the text, but one obvious question arises: why the authors, having in their hands such a vast array of data on HIV genotypes, do not provide the results of the analysis of the distribution of subtypes in the studied population?

First, such data could significantly strengthen the article and provide very important material for other researchers involved in studying the global HIV diversity.

Second, the authors provide data not only on mutations designed to track the transmission of resistant strains, but also on polymorphic mutations, which are very often associated with viral subtype; for example, E138A in RT may be associated with subtype A or C, E157Q in integrase is polymorphic for subtype B, etc. It is not entirely correct to interpret data on polymorphic mutations without defining subtypes, and it may influence the final PDR indicator.

Performing subtype analysis will take some extra effort, but both the article and its readers will benefit.

An additional note to the figures - the resolution is clearly not enough, and the details are small, and it is rather difficult to see everything in the combined figures.

The article is in need of revision, but the information it contains and the way it is presented in general are highly commendable.

Author Response

Reviewer 2

Comments and Suggestions for Authors

In this study the authors present the results of the first large-scale study of HIV pre-treatment drug resistance at the largest clinic in Mexico. The study included 8128 specimens, within 4 years 6688 genotypes were obtained by full genome sequencing and additionally by Sanger sequencing.

PDR to any drug class was 14.4%, with the largest proportion of mutations formed towards NNRTIs. A very detailed and thorough analysis of the data has been carried out in order to identify the groups of the population most at risk of being infected with drug-resistant strains of the virus, and ways to prevent the transmission of resistant viruses have been outlined.

In addition to traditional indicators, social aspects were studied - education, marital status, behavioral features, etc., the relationship between these parameters and the level of PDR was reliably analyzed.

To be true, I have not seen such a careful analysis of the results on this topic before. This data and method of analysis will be very useful for countries, where baseline HIVDR testing is not standard of care according to national guidelines.

There are no special comments on the text, but one obvious question arises: why the authors, having in their hands such a vast array of data on HIV genotypes, do not provide the results of the analysis of the distribution of subtypes in the studied population?

First, such data could significantly strengthen the article and provide very important material for other researchers involved in studying the global HIV diversity.

Second, the authors provide data not only on mutations designed to track the transmission of resistant strains, but also on polymorphic mutations, which are very often associated with viral subtype; for example, E138A in RT may be associated with subtype A or C, E157Q in integrase is polymorphic for subtype B, etc. It is not entirely correct to interpret data on polymorphic mutations without defining subtypes, and it may influence the final PDR indicator.

Performing subtype analysis will take some extra effort, but both the article and its readers will benefit.

A: We thank the Reviewer for the positive feedback and thank them for bringing up the topic of viral diversity and subtype prevalence, which was indeed not analyzed in our study. It has been previously published that the epidemic is largely dominated by subtype B in Mexico and this study group is not the exception. After analyzing HIV subtype, we found that 98.7% of the persons included in the study harbored subtype B viruses. We have now included the variable “HIV subtype” in Table 1 and Table 2 and adapted the text to describe subtype prevalence in the study (results section 2.1). No differences in the prevalence of HIVDR by subtype was observed. Also, possible associations between specific polymorphic mutations and HIV subtypes has been added to the results (Section 2.2).

An additional note to the figures - the resolution is clearly not enough, and the details are small, and it is rather difficult to see everything in the combined figures.

A: We have provided high resolution figures to improve quality.

The article is in need of revision, but the information it contains and the way it is presented in general are highly commendable.

Reviewer 3 Report

The study contains a large number of participants, however the available data for the patients' demographic drastically vary in number of the participants observed in the certain category. Excluding the participants that don't have all the data (for example, patients whose gender is unknown) compared in the table 1. from the study, or at least forming another table that applies only to the certain genders or other categories would not significantly decrease the number of the participant but would, in my opinion add the clarity to the result section. If you are determining the prevalence of pretreatment drug resistance then there is no need for prior and post ARV exposure is unnecessary. Information I believe should be included in the study is the subtype of the sequences obtained from the patients, as well as comparing the SDRM prevalence in different subtypes. Other than that I find the paper quite well written, but the interpretation of the results should be improved before publishing.

Author Response

Reviewer 3

Comments and Suggestions for Authors

The study contains a large number of participants, however the available data for the patients' demographic drastically vary in number of the participants observed in the certain category. Excluding the participants that don't have all the data (for example, patients whose gender is unknown) compared in the table 1. from the study, or at least forming another table that applies only to the certain genders or other categories would not significantly decrease the number of the participant but would, in my opinion add the clarity to the result section.

A: We apologize for this and thank the Reviewer for the excellent suggestion. We have now divided Table 1 into two: a table containing variables collected since 2017 for all participants and a second one including only participants enrolled from June to December 2020 (n=1348), for whom additional variables were collected. The text has been updated accordingly.

If you are determining the prevalence of pretreatment drug resistance then there is no need for prior and post ARV exposure is unnecessary.

A: We understand that ART-naïve and ART-exposed are different epidemiological groups, but think that it is important to analyze them together using the operational definition of “pretreatment drug resistance”, that the WHO has recommended as part of its standardized HIVDR surveillance protocols (https://www.paho.org/en/documents/concept-note-surveillance-hiv-drug-resistance-adults-initiating-antiretroviral-therapy). Inclusion of ART restarters in the surveys is important as they can constitute an important percentage of persons starting first-line ART in some regions of the world. Also, in order for our study to be comparable with others, we believe that including this group is extremely important.

As the Reviewer suggested, we have included a separate analysis of HIVDR levels in ART-naïve and prior ART-exposed persons, using the better described subcohort of participants enrolled from June to December of 2020, for whom the data is available. Stratification of the complete cohort is not possible due to lack of information on the variable of previous ART exposure, especially for participants enrolled early in the study. We have included a new figure contrasting HIVDR levels by drug class in the two groups. We have also updated the text in section 2.2 of the results.

Information I believe should be included in the study is the subtype of the sequences obtained from the patients, as well as comparing the SDRM prevalence in different subtypes. Other than that I find the paper quite well written, but the interpretation of the results should be improved before publishing.

A: We thank the Reviewer for bringing up the topic of viral diversity and subtype prevalence, which was indeed not analyzed in our study. It has been previously published that the epidemic is largely dominated by subtype B in Mexico and this study group is not the exception. After analyzing subtype we found that 98.7% of the persons included in the study harbored subtype B viruses. We have now included the variable “HIV subtype” in Table 1 and Table 2. The text (results, section 2.1) has also been updated accordingly. No differences in the prevalence of HIVDR by subtype was observed. Also, possible associations between specific polymorphic mutations and HIV subtypes has been added to the results (Section 2.2).

Round 2

Reviewer 1 Report

Great job by the authors, all of the issues has been brilliantly addressed.

This paper is ready for publication for me, after adjsuting the size of figure 4 and 5 that seems too wide.

Alessandro

Reviewer 2 Report

Comments have been taken into account, corrections and additions are sufficient, the article is ready for publication.